# The Impact of a Communication Training on the Birth Experience: Qualitative Interviews with Mothers after Giving Birth at Obstetric University Departments in Germany

**DOI:** 10.3390/ijerph191811481

**Published:** 2022-09-13

**Authors:** Martina Schmiedhofer, Christina Derksen, Johanna Elisa Dietl, Freya Haeussler, Reinhard Strametz, Beate Huener, Sonia Lippke

**Affiliations:** 1German Coalition for Patient Safety (Aktionsbündnis Patientensicherheit), 10179 Berlin, Germany; 2Department of Psychology & Methods, Jacobs University Bremen GmbH, 28759 Bremen, Germany; 3Wiesbaden Business School, Rhein Main University of Applied Science, 65183 Wiesbaden, Germany; 4Department of Gynecology and Obstetrics, University Hospital Ulm, 89070 Ulm, Germany

**Keywords:** birth, communication, obstetrics, Health Action Process Approach, healthcare research, obstetric healthcare workers, patient safety, preventable adverse events, qualitative research

## Abstract

(1) Background: Patient safety is a pressing issue in healthcare. Besides economical and organizational issues, human factors play a crucial role in providing safe care. Safe and clear communication on both the healthcare workers’ and patients’ sides contribute to the avoidance of medical errors and increase patients’ and healthcare workers’ satisfaction. Globally, the incidence of experiencing at least one adverse event in obstetrics is about 10%, of which half are classified as preventable. According to international research, improving communication skills may decrease preventable adverse events. The research question was to what extent communication training for pregnant women impacts the quality of communication and mutual understanding during birth. (2) Methods: Communication interventions with pregnant women were conducted in two German university obstetric departments in a mixed methods research design, based on the Health Action Process Approach. The online classes covered the awareness of personal wishes, the understanding and usage of communication strategies, self-efficacy and empathy. This study presents the qualitative results. Out of 142 mothers who answered two questionnaires before the communication training and after the birth, 24 in-depth semistructured interviews were conducted to explore the subjective impact of the communication training. The results were analyzed with qualitative content analysis. (3) Results: The majority of participants felt incentivized to be aware of their personal wishes for birth and to express them. Perceived positive experiences with sufficient competency in communication, empathy and mutual understanding outweighed negative treatments and experiences in the hospital, some of which could be attributed to structural problems. (4) Discussion: The reported positive effects of the communication training underline the need but also the potential for communication lessons to reflect and improve communication skills in obstetrics. However, negative experiences due to structural problems in the healthcare system may be buffered by communication skills but not solved.

## 1. Introduction

Health psychology is the science of human thoughts, emotions and behavior in relation to health and illness. The focus is primarily on identifying risks and developing preventive actions, including psychological and social variables and their interactions with physical illnesses and disabilities. Thus, health psychology is strongly related to and can contribute to enhancing healthcare. Since the Institute of Medicine’s report “To err is human” was published, patient safety has been seen globally as a pressing issue in healthcare. Healthcare systems are environments that are complex organizations whose outcomes are determined by health policies, including budgeting, professional education and regulations, workload, cooperation between in- and outpatient care and the populations’ current burden of diseases [1], which are all crucial risk factors in health and well-being. Accordingly, health psychology can help to develop preventive strategies and thus improve healthcare. The aim of this paper is to scientifically study behavior and mental processes in connection with health and morbidity, psychological processes and psychosocial factors as well as their inter-relation with patient safety and health promotion, as well as to apply behavior change theories specifically to communication in healthcare settings. Accordingly, in the following, we review the state of the art in this regard.

The human factor, which is also a focus of health psychology, plays an important role in providing successful and safe care. To deliver the best possible medical services, healthcare workers (HCWs) must adapt their real-life conditions to the challenges they face in their everyday lives, which is only possible with effective interdisciplinary and patient–provider communication [2,3]. Inadequate patient–provider communication is one of the main causes of patient safety incidences; hence, involving patients in the care process can help to overcome challenges and promote patient safety [1]. Therefore, encouraging and supporting patients in taking an active role in their health behavior and treatment is seen as a key factor in improving health services [4,5].

Patients can contribute via their own health behavior to avoid health hazards and medical errors which cause preventable adverse events (pAEs). pAEs are according to the Harvard Medical Practice Study defined as “unintended injury or complication that results in disability at the time of discharge, death or prolonged hospital stay caused by healthcare management rather than by the patient’s underlying disease process” [6]. pAEs pose a huge burden not only to patients and their families but also to HCWs who are at risk of being affected as well, culminating in being traumatized by causing medical errors, known as the second victim phenomenon [7]. Globally, the incidence of pAEs is high: A recently published scoping review which includes 25 studies across six continents indicates that about 10% of all patients were affected by at least one adverse event (AE), with 7.3% of those being fatal. Importantly, about half of all AEs were classified as preventable (51.2% in a range between 34.3 and 83%) [8]. Strategies to reduce the number of pAEs are implemented worldwide [9]. According to international research, improving communication skills may contribute to decreasing the number of pAEs [10].

The field of obstetrics differs in many aspects from most medical disciplines, which are in charge of patients characterized by their illnesses: The majority of women in obstetrics are healthy and to a certain extent self-determined. They are usually supported and joined by their partners or another socially close person, hence fully capable of voicing their concerns and wishes. Additionally, the time of the pregnancy offers enough time to engage in multiple health behaviors as well as other preventative behaviors. Hence, the pregnancy poses a unique possibility for interventions to empower pregnant women and their partners to avoid health hazards and contribute to their safety by addressing the human factor from a health psychological perspective, e.g., via communication interventions to increase mutual understanding. In this scenario, communication behavior can be seen as an important health behavior and preventative measure. Hence, behavior change theories from the field of health psychology should be considered to improve this behavior. Furthermore, the birthing process is hardly predictable and may change instantly to an emergency situation, making a comprehensive prevention program including good communication even more relevant [11,12].

Medical errors can account for negative outcomes for women and their newborns and lead to high healthcare costs [13]. From that perspective, obstetrics can be seen as a high-risk area in medicine. Not only do long-term somatic impairments or disabilities pose a huge burden to families, but a negative birth perception can also lead to stress and traumatic experiences [14]. For that reason, communication skills are particularly necessary for all stages of birth to ensure safety and maternal satisfaction to prevent the development of diseases [15].

In order to promote women’s and newborns’ safety, several strategies are implemented globally [16]. Of these, studies indicate that communication interventions in obstetrics may contribute to reducing pAEs [17,18]. However, successful communication requires proficient skills on both the HCWs’ as well as the patients’ sides to achieve mutual understanding.

Against this background, we reviewed the existing international literature on communication strategies in obstetrics for successful strategies in a systematic review [19]. However, interventions and outcome measures varied greatly, and the overall level of evidence was low. Only 3 of the included 71 studies targeted mothers (to-be) explicitly in training or intervention approaches [20,21,22]. The effectiveness of these studies also varied considerably. Considering that communication is an important preventive (health) behavior, we decided that communication trainings should be theoretically based on established behavioral theories to target the crucial determinants of safe communication [23]. One behavior change theory that can be applied to communication in healthcare is the Health Action Process Approach (HAPA) [24], a very prominent theory in health psychology. The model assumes that before the intention to communicate safely is formed, individuals (in this case, HCWs and expectant parents) need to be aware of the importance of communication during childbirth and their own risks of not communicating safely. This intention can translate to safe communication behavior via specific planning. Another crucial aspect is self-efficacy: Patients and providers need to be confident to take up and maintain safe communication skills and behavior.

Against this background, communication interventions based on the HAPA model with pregnant women and their partners were conducted in two German university obstetric departments as an example of recent research in health psychology. The overall goal was to improve communication skills and thus increase maternal self-efficacy and satisfaction and decrease the number of pAEs [25].

The study goal was to identify the improvement in communication skills, perceived self-efficacy and mutual understanding with HCWs. We chose a mixed methods approach with a convergent parallel design, that is, quantitative and qualitative data were collected at the same time and analyzed independently. The purpose was to develop a deeper understanding of the research focus by obtaining complementary data. Qualitative data gained through open-ended interviews provide topics not covered by the standardized representative questionnaire; however, quantitative data can help to quantify effect sizes. The data may also include hints for further research [26,27]. The study at hand only presents the results of the qualitative part of the research project, and the results from the quantitative analysis will be published elsewhere.

The research questions were:To what extent does the communication training impact the preparation for childbirth and communication during childbirth from the mothers’ subjective views?How was the quality of communication and mutual understanding in the hospital perceived by the mothers-to-be and their partners?

## 2. Materials and Methods

This research is one module of the “TeamBaby—safe, digitally supported communication in obstetrics” intervention study, which took place at two German universities and teaching hospitals. Obstetric HCWs, mothers-to-be and their partners were trained in assertive communication skills. Detailed information has been described during study registration (NCT03855735) and in the published study protocol [19]. Ethical approval for data collection was granted through the project’s ethical approval from the University Hospital of Frankfurt Medical Research Ethics Committee (Number 19-292) and the University Hospital of Ulm Human Research Ethics Committee (Number 114/19).

At first, HCWs at the study clinics had been trained in safe communication in interprofessional face-to-face formats. The training was conducted by a company run by interprofessional HCWs experienced in obstetrics. The content was developed in close cooperation with the study project team (consisting of health psychologists, masters of science in health promotion and one sociologist) and was theoretically based on the HAPA [28]. For the pregnant women, the same company adapted the training contents and mode of delivery. Due to the COVID-19 pandemic and associated infection control requirements, twenty training sessions were provided in a 2.5 h online format, which was carried out interactively and patient-centered between June 2020 and August 2021. Between 6 and 16 participants took part per training session. To prepare for the training, the participants completed a self-reflection questionnaire to think about their needs and wants for the birth.

The first part of the training was an introduction round to understand the participants’ individual needs and communication approaches. The introduction was followed by interactive tasks. The main topics of the classes for expectant mothers (and partners) were:Risk perceptions and outcome expectancies: awareness about personal wishes for the birthing process and the role of communication.Intention: understanding communication strategies (four sides of communication—communication square) [29].Planning and self-efficacy: expressing personal needs and wishes, inquiring (close-the-loop) and speaking up.Support: building empathy for professionals by taking their perspectives (empathy maps) and mobilizing/accepting support.

A detailed overview of the educational program and the training content can be found in other publications [27,30]. Finally, participants were invited to practice their communication strategies and learned skills in a 1:1 interaction with one of the trainers (an experienced midwife) with examples from obstetric care and emphasizing their individual needs from the preparation questionnaire.

A total of 255 mothers and 125 fathers-to-be who planned to give birth at one of the study clinics took part in the communication training consisting of lessons about two to ten weeks before the expected date of birth (see Figure 1).

Enrollment occurred via the study clinics’ homepages, social media and personal approaches by study personnel. Before the training and after giving birth, a total of 142 mothers in the intervention group answered two questionnaires. The quantitative evaluation regarding the questionnaire data is currently under review [30]. To gain a better understanding of the standardized responses, we wanted to conduct complementary qualitative interviews with participants in the intervention group. The goal was to obtain a broad picture of women with different backgrounds regarding age, migration history, mode of delivery, medical complications during birth such as (emergency) C-sections, study clinics and COVID-19 infection status. A stratified purposive subsample of 30 mothers with broad heterogeneity was approached to participate. The preselection was conducted by the study staff at the clinics, who had access to the demographic and clinical data within the scope of data protection. Potential participants were targeted by respective characteristics such as age groups or birth modus in order to obtain the broadness of the sample. Two of the contacted persons declined and four did not answer the email. Between February and July 2021, one author conducted 24 in-depth semistructured interviews in a period from 4 to 33 weeks after birth. Recruitment ended when data saturation was reached and no new findings emerged [31]. Sociodemographic information of the participating mothers-to-be is displayed in Table 1. The semistructured interview guide (Table 2) includes the main topics to stimulate a detailed description of the birth experience with a focus on communication, perceived support and the subjective impact of the communication training. It was flexibly adapted to the narrative flow and developed on the basis of our previous research results and project planning [19,25,32,33].

### Data Analysis

Before starting the interviews, all participants gave written informed consent. After the interview, fieldnotes were taken to document nonverbal communication, atmosphere, the quality of internet connection and other features. All interviews were audio-taped and transcribed verbatim by students at Jacobs University Bremen. Word-for-word transcripts and fieldnotes were entered anonymized into the qualitative data software MAXQDA2020. The results were analyzed in a multistage process by means of qualitative content analysis (QCA). The QCA approach codes inductively and deductively according to themes emerging from the data analysis [34,35].

In the first step, one author (MS) reviewed the transcripts line by line. Interview sequences or single words were assigned to broad categories.

Once the interviews had been transcribed verbatim, the research group (JD, FH, CD, and MS; SL), including health psychologists, a master of science in health promotion and a sociologist, worked out the main categories to answer the research questions. Several discussions were held until a consensus was reached for all categories. The results are presented in the next section and illustrated with original quotes. After comprehensive discussions within the research group (JD, FH, CD, and MS), data related to the COVID-19 pandemic are published in a distinct publication [36].

The results were structured according to the research question, starting with the subjective impact of the training lessons on personal preparedness and communication competence for childbirth. This was followed by the perception of the quality of the communication in the hospital, covering positive and negative experiences concerning mutual understanding, interpersonal adaption, empathy and communication competence. Data interpretation was based on the subjective views of the interviewees in compliance with QCA. The main topics are presented in detail according to this structure and are illustrated by meaningful quotations in the result section.

## 3. Results

Overall, the majority of interviewees appreciated the offer of the communication trainings for both the HCWs as well as the expectant parents as a beneficial commitment of the study clinics. Given the advanced stages of pregnancy, the convenient participating mode from home was valued. In addition, the benefit of writing questions into the chat of the online bar instead of taking the floor in an unfamiliar group was mentioned positively. The advantages of the online conduction outweighed the disadvantages of less face-to-face contact.

### 3.1. Impact of the Training on Preparation for Childbirth and Communication during Childbirth

Almost all respondents reported that the training provided the option to think intensively about the upcoming birth process, which was beneficial in terms of improving their subjective outcome expectations and their personal contribution to reaching those expected outcomes. This positive impact was also stressed by participants who were already familiar with the communication models presented:


*“So I have dealt with it (senders and receivers) quite intensively during my studies. And even for me, I thought at during this training, ah, yes, that was true. And I don’t think I would have thought about it in advance, without that. So that’s our (partner included) opinion, it would probably be very, very useful for every expectant parent to do such a training”*
(P_21).

The trained women varied in age and personality, parity status and the risk of medical complications during pregnancy or birth. The guided introduction round displayed the range of expectations and backgrounds of participants. At the same time, it mirrored the challenges the HCWs were supposed to meet.


*“This exchange with the other pregnant women and the course instructor was important because you could see that other women wanted completely different things for the birth than I did (…). Other mothers have said, I want to be left alone as much as possible. Then the thought came to me, ‘YES; how should the midwife know that? How should she know my wishes if I never express them?’ And then it occurred to me that I might have to communicate more”*
(P_08).

The pregnant women appreciated the inclusion of their partners in the training session since the communication within the family became a topic that could be more openly discussed. Additionally, partners were seen as an important and integral part of the birthing process and the training helped them to prepare for their role. Frequently, the training initiated an intensive discussion with the partner about their respective expectations and responsibilities during the birth:


*“For us, this training was actually a start to prepare the birth more intensively. And we just took that as a hanger, okay, we have to know exactly what we want, that we can communicate what we want. And then I started to talk very clearly with my husband. These are my worries, these are my fears”*
(P_21).

Although some women reported that labor pain and the rapid pace of birthing resulted in feeling an inability to think or communicate clearly, most of the training participants were able to apply some of the training content or at least felt more confident during the birthing process in hospital.


*“I was so afraid of the pain (…) I then learned during communication training that I should express my feelings beforehand. That’s why I then said, I’m so afraid of pain, please help me have less pain during birth. Exactly. That was very good”*
(P_22).

### 3.2. Perceived Quality of Communication and Understanding in the Hospital

The application and practical usage of the training contents corresponded with the duration of the birthing process and the length of stay in the hospital. Furthermore, the interviewees reported different experiences with personal behaviors and organizational structures. All study participants knew beforehand that the obstetric staff had already been trained in safe communication in the course of the TeamBaby project and thus expected good communication skills. Even though these were met in many cases, burdening situations with a perceived lack of understanding were reported. Respondents distinguished between the individuals by whom they received care, professional groups and care settings.

Midwives were described as the ones giving the most emotional support during the process of the birthing [37]. They spent more time with the women and built relationships. However, the leading role of physicians was appreciated in medically challenging situations.


*“Before the birth, I may have wanted a doctor, but that was not necessary for us and that was okay, but I think I would have felt even safer with the doctor. I don’t doubt the competence of midwives either. But maybe it is also because in (participant’s country of origin) it is usual that the doctor is present at the birth”*
(P_16).

Additionally, the midwife was seen as a “mediator” between the doctors and the pregnant women, especially in case of unexpected C-sections:


*“And she was also a bit of a mediator, I suppose, between the doctors and me, in case something had happened. She would probably have said, I don’t think the patient is doing so well or something. Whereas the gynecologist, she didn’t see me at all, I was already covered”*
(P_05).

Despite these differences that were perceived between medical professions, the interviewees reported that persons from the same profession had different communication approaches. These were often attributed to (medical) experience that the women had trust in, as the following example illustrates:


*“When the water broke at some point during the birth, the amniotic fluid was quite dark green and then the senior physician was immediately called in and she also stayed there the whole time, that was, so nothing against the resident, but the senior physician has even more experience, she also brought more calm into the whole thing, I found that very pleasant”*
(P_01).

On the other hand, the women valued personal engagement regardless of the professional experience and felt reassured even if a student tried to meet their needs:


*“So as a positive example, a midwife student comes to mind. How patiently she listened to me every time I had questions. And she didn’t make me feel like I was asking stupid questions. (…) Even if she didn’t know something, she said with so much respect: ‘Mrs. (name), I don’t know, but please give me two hours. I will clarify this with the experienced colleague and then I will come back.’ Two hours later she was back and provided me with an answer”*
(P_06).

Regarding the understanding of organizational problems and staff shortage, the women explained that the training changed their perception of the HCWs’ roles and their communication. The training focused on mutual understanding and perspective taking, which the women reported as helpful.


*“What I found particularly good about the training was that we put ourselves in the shoes of the professionals. (...) Because when you were in the delivery room, you remembered that, ok, they have a lot to do right now and for you it’s such an exceptional situation, but for them it’s everyday life, what they’re doing right now. And that was, especially in terms of communication, that you weren’t too demanding, but rather took into account a little bit, ok, let them do their rounds now and if you have another detailed question, then they will take the time, if they have it. And that has already helped me. Absolutely”*
(P_10).

On the other hand, the empathy for the staff taught in the course even led some participants to regret not having been able to communicate well enough with the HCWs during their stay in the hospital.

### 3.3. Positive Experiences with Sufficient Communication Competence and Perceived Empathy

Interviewees reported a number of positive experiences that they attributed to the implementation of communication training for both HCWs and expectant mothers. The first positive experience centered around being taken seriously. The women felt that, especially during the birth process, the midwives and physicians respected their bodily autonomy and decisions. In some cases, the parents-to-be took written notes of their needs which were perceived and respected:


*“We noticed exactly, all have read this sheet (questionnaire regarding wishes for birth). Even the midwives, they knew exactly what was going on. Things like ‘we want to take our placenta home with us’ or ‘we don’t want an epidural at first’. I wasn’t offered an epidural until I asked for one. And then I realized that the doctors and midwives had read it. Or one of them had read it and told the others”*
(P_03).

With some staff members, especially midwives and students, the women felt a personal connection. This was based mostly on the personal interest of HCWs who spent more time with the mother, which was described as very positive environmental aspect.


*“And then a really great midwife team came. That was actually the most beautiful phase of the birth. That was a trainee and an instructor. And they did a really, really good job. They explained everything in detail. And you also noticed that they had time. They spent quite a lot of time in the room with me. (…) One of them, the student, came two days later to the maternity ward and asked how we were doing. Visited us”*
(P_17).

Particularly in challenging situations, reassuring communication was highly valued. This included personal support, for example, if a decision about a C-section had to be made, but also the time in the maternity ward, where women were cared for after a C-section.


*“A psychologist was also there one day later. She talked with me for half an hour. This was ordered by the doctor who operated on me. I also thought that was great. And the doctor was there again on the next day. She told me that it was a good decision (C-section). She didn’t want me to go out with a trauma. (...) She really took her time again”*
(P_30).

In the end, perceived interpersonal adaption and perceived empathy were reported to have positive impacts on the trust in HCWs and the perception of a high quality of care.


*“And my midwife took over a lot of this prepartum talk, so I had the wish for a birthing stool. In the end, I didn’t use it. But she had already set everything up. And I had checked fragrance would be nice. Then she asked me which fragrance I would like and then she added it (...)So she was not the midwife with whom I had spoken before. But she knew the file. And I just felt like I was in such good hands. She was also there the whole time, because I had told her, I’m always afraid when no one is there”*
(P_08).

### 3.4. Negative Experiences with Insufficient Communication and a Lack of Interpersonal Adaption and Empathy

In contrast to the positive experiences that were described above, the communication training could not avoid all negative experiences. Especially on an environmental, i.e., organizational, level, there remained some issues that the women reported as stressful.


*“What I found problematic was that some nurses still went through their program at night. It’s 11:30 p.m. and you’ve just gone to sleep for the first time, your baby doesn’t want to be breastfed and you’re glad you can close your eyes. Then the nurse comes in, turns on all the lights, cheerfully says “Hello”, wakes everyone up and says “So, we’re going to do the Hessel Screening” or whatever there is. We’re going to prick your child’s heel now or we’re going to do a hearing test or things like that, which I would have expected in the morning”*
(P_03).

Organizational problems seemed to lead to delayed treatment in some cases. Although the participants tried to use their resources to communicate their needs, it sometimes took a long time until they were met. This was particularly the case if other disciplines had to be involved, for example, if epidural anesthesia was required.


*“And then it was so intense that I said, now I need an epidural. Because I can’t stand it for that long again. And then it took an insanely long time. At first no one came. There was no real information about how long it would take. So I would have liked to know a time horizon. ‘You have to hold out for about half an hour’ or something like that. I then sent my husband there several times and said, go and have another look. Let me know, somehow. Because that is now really unbearable”*
(P_14).

In some cases, the communication seemed to fail and the mothers felt like they missed crucial information. In one example, the mother was not aware that her child had high levels of bilirubin until an intervention was required, which initially scared her, since she had no information about this before:


*“I had been a mother for 20 h and I have no idea about diseases as far as children are concerned and blood values. I’m not from that area. And the nurse comes in the second morning. And tells me, ‘Mrs. (name), the bili has gone up.’ And I look at her and I say, ‘What?’ ‘Yes, the bili has risen, the child has to go under the lamp’”*
(P_06).

During the birth or the phase before delivery, the mothers reported a number of situations in which the communication was perceived as insufficient. This was related to a lack of both interpersonal connection and information:


*“So the resident came and listed the factors that are not so optimal and that they also discuss this again with the leading physician. In retrospect, I thought that at that moment I knew what she wanted to tell me. She had already listed the issues that were in favor of a C-section, but she hadn’t made it clear at that moment”*
(P_04).

Although it was conveyed in the trainings that problems arise from misunderstandings between HCW and patient, some women blamed themselves:


*“I had these really bad contractions after the induction and in retrospect I think to myself, yes, you have pain, but it doesn’t have to be that bad and I should have gotten help beforehand. I should have simply let them know beforehand. But then I was shy because the midwife said at the beginning, ‘Well, I have to stay on the ward, I couldn’t go down to the delivery room’”*
(P_37).

In one instance, the mother reported feeling overwhelmed and exhausted after a long birth process which was worsened by a perceived lack of empathy and interpersonal adaption of the staff.


*“My hands were shaking and I was somehow not really on-top. You need a little sleep after four days. And she (midwife) noticed how I was feeling and she said, it’s no wonder that my child cries so much when I’m in such a bad mood (...) I think if you’re overwhelmed, you should try to communicate differently. Or maybe ask for understanding that she now has to do everything quickly because they are understaffed or whatever. I have understanding for something like that, if someone tells me that, but if someone is simply unfriendly with me, I don’t understand that”*
(P_06).

Taken together, Figure 2 displays the main results, which are discussed in the next section.

## 4. Discussion

The aim of the study at hand was to examine the subjectively perceived impact of participation in a communication training for expectant mothers (and fathers) on their birth experience to overcome how in the healthcare system there is still less focus on maintaining health and more on treating illness. This recent research in health psychology is embedded in the TeamBaby study with the overarching goal of enhancing patient safety through HCWs’ and expectant parents’ improved communication skills. The data surveyed with a questionnaire are subjects of other publications [25] which are currently under review [30,38].

The goal of the qualitative study part was to explore the perspectives of a subset of all study participants to obtain a deeper understanding of how the communication training influenced their birth experience. For that reason, semistructured interviews were conducted in which the participants narrated openly about their impressions.

### 4.1. Impact of the Training on Preparedness for Childbirth and Communication during Childbirth

Almost all participants considered the communication training as helpful and encouraging for reflecting on and expressing their personal needs, which confirms previous research on interventions to enhance patient–provider communication [39]. Even interviewees who were educated about communication models before emphasized the effect of a heightened awareness towards the counterpart of an interaction. They recognized that HCWs cannot know about their personal expectations if they do not voice them, hence developing a risk awareness that can set the stage for a change in communication behavior [24].

Another reported benefit was the intensive exchange with the partner about mutual expectations and the respective roles during labor. Additionally, women reported that they prepared and planned their communication, e.g., by providing the HCWs with written wishes for the birth process. The online format chosen due to the COVID-19 pandemic was seen as an advantage over a face-to-face event. This result is an important indication for the future implementation of communication trainings conducted in non-pandemic times. Many women valued the convenient participation from home and some highlighted the easy access to the discussion bar in case of upcoming questions. This is in line with research that was conducted during the pandemic, showing that digital interventions are acceptable and feasible [40].

Our qualitative results are in agreement with international research which shows that successful communication during childbirth requires the active participation of pregnant women. Furthermore, the results are in line with the quantitative study that evaluated the questionnaire data from the recruitment sample compared to a control group who received care as usual. In the manuscript which is currently under review, quantitative analyses reveal a positive effect of the intervention on communication behavior, perceived safety and birth outcomes as well as HAPA variables including planning and coping self-efficacy [30]. This points to the general effectiveness and thus importance of training, as the systematic consideration of one’s communication behavior does not seem to be common [41].

In this study, the reported positive effects were already achieved by a 2.5 h one-time offer, even though communication models could neither be introduced nor practiced in depth. This underlines the need but also potential for communication lessons to reflect and improve communication skills. To deepen and to strengthen the contents and to offer flexible and convenient tools, online formats can be provided, for example, as an asynchronous online training in the form of a Massive Open Online Course (MOOC) or an app [19].

Most of our interviewees were able to practice parts of the training contents, even though labor pain sometimes kept them from expressing themselves clearly. The mutual understanding of the demands and duties of HCWs and expectant parents was one part of the communication lessons by providing insights into working procedures and demands. Indeed, some respondents reported how their understanding of the everyday requirements of the HCWs eased waiting times or uncertainties.

However, some women were sorry about not having shown sufficient empathy towards the HCWs, which might have caused some pressure. Previous research has determined that the mutual understanding between patients and HCWs can positively influence patient outcomes due to patient empowerment, therapeutic alliances and higher-quality medical decisions [42]. This illustrates the link between how effective communication is related to the avoidance of pAEs and thus an increase in patient safety. The same applies to problems in patient–provider communication [43]. Another potential positive effect is that a reduction in adverse events due to patient involvement and effective communication can prevent HCWs from becoming second victims. By reducing pAEs, not only patients but also HCWs can avoid experiencing anxiety, frustration up to severe depression and PTSD [44].

However, this perspective is in part an unplanned outcome of the communication lessons, as it resulted unintentionally in blaming themselves for not expressing their needs sufficiently. It seemed like those women felt less prepared for communication in difficult situations, mapping back to the constructs of coping self-efficacy and coping planning which is crucial for behavior change and classical health promotion [24].

### 4.2. Perceived Roles of Professionals and Commitments during Childbirth

Expectations regarding the care received from different occupations and professional levels may differ, as some participants expressed. Midwives are described as the ones giving the most emotional support during the process of birthing, which is consistent with their professional roles [45]. However, the intensity and quality of care received by physicians or midwives may vary due to medical complications. Interestingly, midwives were reported as “mediators” between interviewed patients and partners and doctors, while the latter provided a higher perception of medical safety. Those results underline the need for good interprofessional cooperation in interprofessional teams [32]. Additionally, the expectant mothers valued the involvement of trainees who acted as additional team members and were very approachable. This was in line with previous research on involving students in obstetric care [46].

### 4.3. Positive and Negative Experiences with the Quality of Communication

The narratives about the quality of communication in the hospitals cover a wide range of positive experiences concerning empathy, mutual understanding and the commitment of the HCWs. The data interpretation must consider that the training might have raised the sensitivity concerning communication skills such as a classical health promotion and preventive intervention. To date, this has only been found in communication trainings for HCWs who initially tend to overestimate their knowledge and skills regarding communication [47]. In terms of the HAPA model, the awareness of being at risk of not behaving adaptively and expectancies of being able to reach a certain outcome have to be developed before an intention can be formed to improve behavior [28].

A major positive experience was the perception of being taken seriously. In accordance with the lessons learned in the communication training, many respondents emphasized the respect for their bodily autonomy and the acceptance of personal wishes by HCWs. Even further, the perception of empathy and interpersonal adaption was highly valued when the personnel supported and comforted the women in challenging situations, which is in line with previous research [48].

However, contrasting experiences were also reported. They were often related to environmental, i.e., organizational, difficulties, such as routine examinations at night or delayed care due to a lack of personnel. It must be taken into account that obstetrics, like many inpatient areas, suffers from staff shortages and economic strains which lead to decreased patient satisfaction [49]. In this light, sufficient and good communication can promote empathy and an understanding of waiting times or inconveniences but does not solve such challenges and burdens. COVID-19 containment rules and staff shortages may have led to an aggravation of insufficient resources, as we reported in an associated paper [36].

Criticism about unclear information, such as uncertainty about the progress of the birth, a lack of important information for decision making or unsettling statements, was also voiced. A recently published study emphasized the importance of communication during birth and the postpartum stage to improve women’s satisfaction [33]. In this study, the provision of clear information by the HCWs was seen as the most helpful aspect of verbal communication [50].

On the positive side, these negative experiences reported by our study participants worked as incentives in some cases to speak up or persistently ask for sufficient information. That is, participants took responsibility for their well-being by insisting on safe communication, as it was introduced in the communication training and reflects the component of self-efficacy [51]. On the negative side, the training unintentionally led some respondents to blame themselves for not adequately expressing their needs.

### 4.4. Limitations

Future researchers and practitioners need to consider several limitations of this study when applying its results. First of all, qualitative analysis is always subjective. Although measures were undertaken to reduce interview bias, it cannot be completely excluded. Furthermore, the sample included only participants who gave birth at two university obstetric departments, so the results may differ from populations of non-university clinics.

The participants’ perceptions were not qualitatively compared to those of the women who did not receive the communication training at the same time period. However, the study regarding quantitative data reveals significant interaction effects. It must be noted that these quantitative results have not yet been subject to peer review. In addition, this study reflects only the maternal perception of the communication and does not take the HCWs’ perspectives into account [30]. Therefore, the generalizability of the results is limited, even though the results are in line with previous research results.

Authors should discuss the results and how they can be interpreted from the perspective of previous studies and of the working hypotheses. The findings and their implications should be discussed in the broadest context possible. Future research directions may also be highlighted.

## 5. Conclusions

The study at hand reflects the subjective results of a communication training for (expectant) mothers on their birth experience based on the Health Action Process Approach (HAPA), a prominent theory in health psychology. It presents promising effects concerning the expression of personal needs, mutual understanding and an increased awareness of safe communication as an impact of the training. It is known that a positive birth experience has a sustained impact on the health and quality of life of both mother and child [14]. Health psychology follows a biopsychosocial rather than a biomedical model. This means that special attention is paid to psychological and social factors and their interactions with health and illness. The current study is an example of very recent research in which the HAPA was applied to communication behavior, and the data demonstrate that this is a very valuable approach.

While giving birth is a unique or in any case a special event, HCWs are faced with professional challenges every day. For them, clear communication and mutual understanding are not only essential for increased job satisfaction but also measures to avoid pAEs, which in turn may lead to a lower rate of second victims [7,52] or human diseases, especially the illnesses that are induced by environmental exposure to health hazards, such as human failure. Rather, resources for improved communication skills and support in implementing them in working routines could improve job satisfaction and contribute to a reduction in the costs related to AEs [25,30,53].

Given the fact that obstetric care is faced with everyday obstacles such as shortages of staff, communication skills are a facilitator in insufficient working conditions. More research is needed to confirm the results of our study and to further improve the outcomes of communication training. In addition to the HAPA, there are also extended theories in health psychology and beyond which have been applied to pregnancy [54] and could be researched with regard to communication and birthing in the future, too. The reported positive effects of the communication training underline the need but also the potential for communication lessons to reflect and improve communication skills in obstetrics.

Our research results indicate that the healthcare system would benefit from clear communication and mutual understanding, both in terms of satisfaction and reduction in preventable adverse events. Thus, we assume that it is beneficial and necessary to align approaches to train good communication. Therefore, we recommend the introduction of communication trainings for both sides, healthcare workers and patients, respective expectant mothers, on a regular basis. Adding the important topic of communication to antenatal preparation classes could help to improve patient safety and satisfaction with birth. This is especially important for countries in which obstetric care needs to be further developed; however, the training approach at hand clearly needs to be adapted and re-evaluated in different cultural settings and in different languages, including countries with lower patient safety or issues regarding fertility rates. It is crucial to ensure that especially women who have language barriers or lower communication skills and who are thus are at risk of not communicating safely can benefit from communication interventions to ensure safe childbirth and the well-being of mothers as well as their offspring. Accordingly, communication training is recommended for different target groups and for improving diverse proximal or distal consequences.

## Figures and Tables

**Figure 1 ijerph-19-11481-f001:**
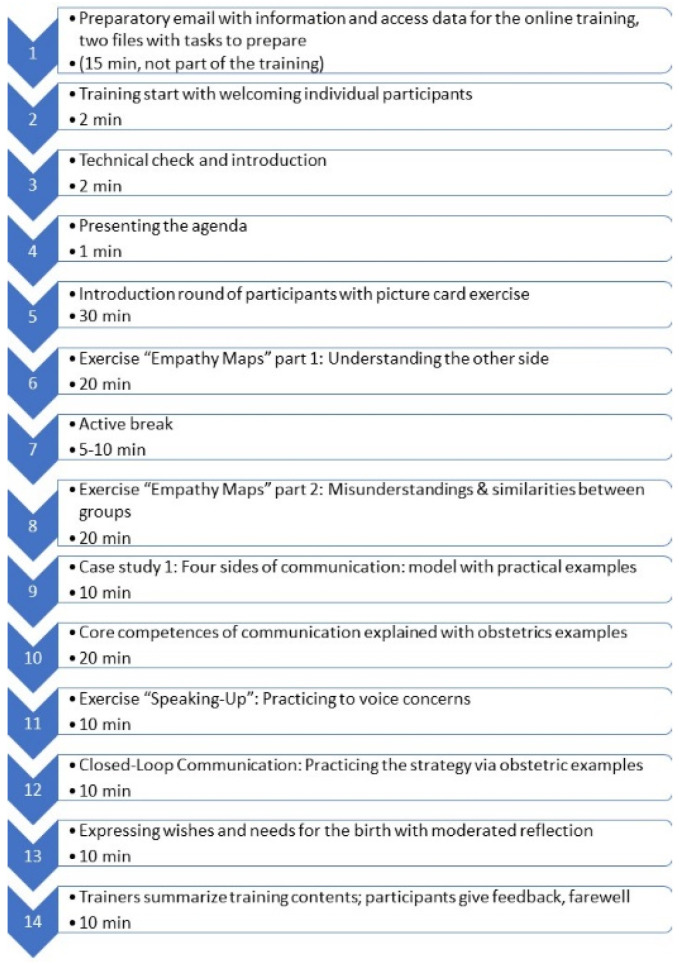
Overview of the content of the communication training. For details, please see [27,30].

**Figure 2 ijerph-19-11481-f002:**
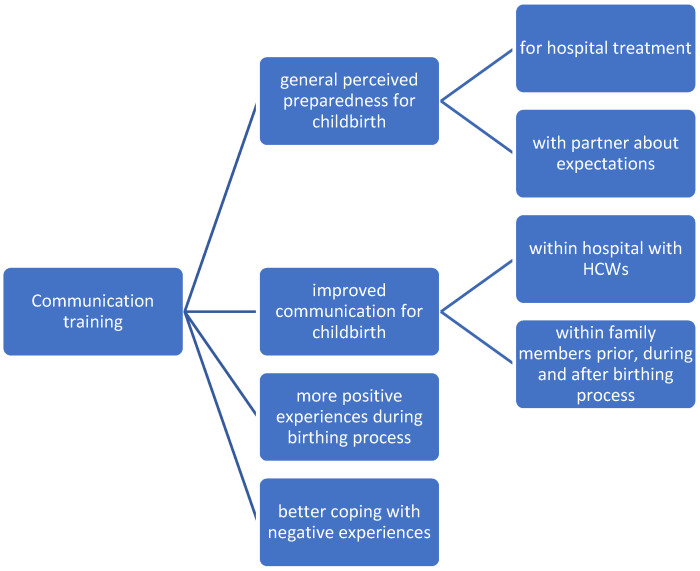
Overview of the reported results of the communication training.

**Table 1 ijerph-19-11481-t001:** Demographics of participants *.

**Age of the study participant**	range = 27–46 years; median 33 years; mean 34 years (SD = 4)
**Migration status**	5 (21%), of which: 3 (12.5%) < 10 years immigrated themselves and 2 (8.5%) second generation
**Parity**	18 (75%) first-time mothers 6 (25%) second and multiple birth
**Birth mode**	18 (75%) vaginal 6 (25%) C-section
**Twin birth**	1 (4%)
**Child with disability**	1 (4%)
**COVID-19 tested positive**	2 (8%)
**Interview duration**	21–95 min; median 37; mean 36 min (SD = 15)
**Interview format**	21 (88%) via TEAMS with camera on; 3 (12%) by telephone/TEAMS camera off
**Interview period**	February to July 2021
**Period of giving birth**	July 2020 to May 2021
**Interview time after delivery**	3.5 to 33 weeks; median 11; mean 13 weeks (SD = 8)

* Data collected from medical records and/or interviewees.

**Table 2 ijerph-19-11481-t002:** Semistructured interview guide (main topics).

Do you remember getting your first information about births in your life, and if yes from whom? Do you recall an emotional association to your recent birth experience—what describes the feeling you associate with your birth the best? How and with whom did you prepare for your birth—giving process? (doctors, midwives, nurses, friends, relatives)
You attended an online communication training at (date). If you recall, what impact did the training have on your preparation for the delivery?
What kind of thoughts have you given to the delivery process? To what extent have you considered how to articulate your needs?
Now it’s about your recent birth experience: Could you please recall the whole process of delivery from the moment you decided you have to go to the hospital to the time of discharge?
What kind of support did you receive, what was sufficient, what was lacking, how did you feel?
How did you perceive the communication with the professionals? Were your questions answered?
How far do you perceive your mental preparation as helpful? How would you express your needs?
Which persons or professions (midwives, doctors, nurses, partners) were most important in providing or lacking support
What could have been better? And who should have done something differently?

## Data Availability

Original data are not available for data protection reasons.

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
