# Peer review of "The Impact of a Communication Training on the Birth Experience: Qualitative Interviews with Mothers after Giving Birth at Obstetric University Departments in Germany"

_ijerph, 2022, doi:10.3390/ijerph191811481_

Round 1

Reviewer 1 Report

First of all, I am very happy to review your thesis. Good luck. However, some suggestions are made for further research.

1. Please present the theoretical background that is the background of your research in detail.

2. You need to explain why you chose the mixed study method.

3. It is necessary to explain the mixed research method in detail. In addition, the rationale for calculating the subject should be presented.

4. The development and contents of the educational program should be presented in detail.

5. It would be good if your research suggests a country that is suffering from problems such as low fertility or implications for policy resolution.

Author Response

Reviewer 1: First of all, I am very happy to review your thesis. Good luck. However, some suggestions are made for further research.

Response: Thank you for your kind comment! In the following, we comply with your suggestions applying them to improve the manuscript. Please find below our responses.

Reviewer 1: 1. Please present the theoretical background that is the background of your research in detail.

Response: Thank you for this comment. In the introduction, we explained the goal of health psychology to identify risks for health-impairing behaviors and to develop and test interventions to improve them. In our study, we wanted to scientifically study behavior and mental processes in connection with health and morbidity, psychological processes and psycho-social factors as well as their interrelation with patient safety and health promotion, as well as to apply behavior change theories specifically to communication in healthcare settings. The rationale is that inadequate and poor communication plays a significant role in triggering adverse events in healthcare, which are common in developed and developing countries. We decided to test the effect of communication trainings in obstetrics because of several characteristics of this subject: the treated persons are usually not typical patients, but healthy women who are fully capable in expressing their wishes and needs during childbirth which is a unique chance for patient empowerment. In addition, health damages can affect both the mother and newborn and lead to life-long and costly burdens which affect the whole family.

To get an overview of conducted communication trainings in obstetrics, we reviewed international communication interventions in obstetrics. Even though several studies indicated that communication interventions in obstetrics may contribute to reducing preventable adverse events, our systematic review revealed a need for further research in addressing mothers. Against this background, we decided to develop communication trainings for mothers-to-be and to apply the established Health Action Process Approach to test for the effects of the communication trainings.

To make our theoretical background and this rationale more transparent, we made changes to the introduction section. Please see pages 1 to 3 and our highlighted changes in the revised manuscript.

Reviewer 1: 2. You need to explain why you chose the mixed study method.

Response: Thank you for this comment. We added an explanation why we chose a mixed methods approach and included two references.

It now reads: “The study goal was to identify the improvement in communication skills, per-ceived self-efficacy and mutual understanding with HCW. We chose a mixed methods approach with a convergent parallel design, that is quantitative and qualitative data were collected at the same time and analyzed independently. The purpose was to de-velop a deeper understanding of the research focus by obtaining complementary data. Qualitative data gained through open-ended interviews provide topics not covered by the standardized representative questionnaire; however, quantitative data can help to quantify effect sizes. The data may also include hints for further research [25, 26]. The study at hand only presents the results of the qualitative part of the research project, the results from the quantitative analysis will be published elsewhere.” (with references 25.       Dixon-Woods, M.; Agarwal, S.; Jones, D.; Young, B.; Sutton, A., Synthesising qualitative and quantitative evidence: a review of possible methods. J Health Serv Res Policy 2005, 10, (1), 45-53., and 26. Guest, G.; Fleming, P., Public Health Research Methods. In SAGE Publications, Inc.: 55 City Road

55 City Road, London, 2015.).

Reviewer 1: 3. It is necessary to explain the mixed research method in detail. In addition, the rationale for calculating the subject should be presented.

Response: Thank you for this comment. We added a detailed description of our sampling strategy.

It now reads: “To gain a better understanding of the standardized responses, we wanted to conduct complementary qualitative interviews with participants in the intervention group. The goal was to obtain a broad picture of women with different backgrounds regarding age, migration history, mode of delivery, medical complications during birth such as (emergency) C-sections, study clinics and COVID-19 infection status. A stratified pur-posive subsample of 30 mothers with broad heterogeneity was approached to partici-pate. The pre-selection was conducted by the study staff at the clinics, who had access to the demographic and clinical data within the scope of data protection. Potential participants were targeted by respective characteristics such as age groups or birth modus in order to obtain the broadness of the sample. Two of the contacted persons declined and four did not answer the email. Between February and July 2021, one au-thor conducted 24 in-depth semi-structured interviews in a period from four to 33 weeks after birth. Recruitment ended when data saturation was reached and no new findings emerged [31]. (31. Saunders, B.; Sim, J.; Kingstone, T.; Baker, S.; Waterfield, J.; Bartlam, B.; Burroughs, H.; Jinks, C., Saturation in qualitative research: exploring its conceptualization and operationalization. Quality & quantity 2018, 52, (4), 1893-1907).

Reviewer 1: 4. The development and contents of the educational program should be presented in detail.

Response: Thank you for this comment. To provide a detailed overview of the educational program we added a figure with the training content and made a reference to the papers in which we describe the content more extensively. As the paper with the quantitative evaluation is currently under review and not published so far, we cannot give the full table but refer to in.

Additionally, we have added some more details to the text which now reads: “For the pregnant women, the same company adapted the training contents and mode of delivery. Due to the COVID-19 pandemic and associated infection control requirements, twenty training sessions were provided in a 2.5 hours online format, which was carried out interactively and patient-centered between June 2020 to August 2021 (see Figure 0). Between 6 and 16 participants took part per training session. To prepare for the training, the participants a self-reflection questionnaire to think about their needs and wants for the birth.

The first part of the training was an introduction round to understand the participants’ individual needs and communication approaches. The introduction was followed by interactive tasks. Main topics of the classes for expectant mothers (and partners) were:

  • Risk perceptions and outcome expectancies: Awareness about personal wishes for the birthing process and the role of communication
  • Intention: Understanding of communication strategies (Four sides of communication - Communication Square) [29]
  • Planning and self-efficacy: Expressing of personal needs and wishes, inquiring (close-the-loop), speaking-up
  • Support: Building empathy for professionals by taking their perspective (Empathy maps) and mobilizing/accepting support

A detailed overview of the educational program and the training content can be found in other publications [27, 30]. Finally, participants were invited to practice their communication strategies and learned skills in a 1:1 interaction with one of the trainers (an experienced midwife) with examples from obstetric care and emphasizing their individual needs from the preparation questionnaire.”

Reviewer 1: 5. It would be good if your research suggests a country that is suffering from problems such as low fertility or implications for policy resolution.

Response: Thank you for this comment. Indeed, Germany as many other developed countries suffers from a fertility rate below two children per woman. Although main reasons for low birth rates are thought to be social and economic, of course a negative birth experience may keep a woman from having a second child, too. Fertility rates are may also be affected by better communication and the effects of a communication training. In the conclusions section, we added three sentences with policy suggestions.

It reads: “Our research results indicate that the healthcare system would benefit from clear communication and mutual understanding, both in terms of satisfaction and reduction of preventable adverse events. Thus, we assume that it is beneficial and necessary to align approaches to train good communication. Therefore, we recommend the intro-duction of communication trainings for both sides, healthcare workers and patients, respective expectant mothers on a regular basis. Adding the important topic of communication to antenatal preparation classes could help to improve patient safety and satisfaction with birth. This is especially important for countries in which obstetric care needs to be further developed; however, the training approach at hand clearly needs to be adapted and re-evaluated in different cultural settings and in different languages, including countries with lower patient safety or issues regarding fertility rates. It is crucial to ensure that especially women who have language barriers or lower communication skills and who are thus are at risk of not communicating safely can benefit from communication interventions to ensure safe childbirth and well-being of mothers as well as their offspring. Accordingly, communication training is recommended for different target groups and for improving diverse proximal or distal con-sequence.”

Reviewer 2 Report

Dear authors, thank you for giving me the opportunity to review your manuscript: “The impact of a communication training on the birth experience: qualitative interviews with mothers after giving birth at obstetric university departments in Germany”. 

I think the article is well written, the results are precise, I did not identify ethical issues, it presents results of interest, and the references are relatively recent.

I send below some comments to improve the manuscript.

Introduction:

-        Line 75-79: Please, provide references.

Conclusion

Authors should add future directions. It also seemed important to me that the authors explore specific future directions, given their results on 'communication training.’

Author Response

Response: Thank you for the nice comment to our manuscript. Please find below our responses.

Reviewer 2: Introduction:

Line 75-79: Please, provide references.

Response: Thank you for your comment. We added two references:

Aibar, L.; Rabanaque, M. J.; Aibar, C.; Aranaz, J. M.; Mozas, J., Patient safety and adverse events related with obstetric care. Arch Gynecol Obstet 2015, 291, (4), 825-30.

Panagioti, M.; Khan, K.; Keers, R. N.; Abuzour, A.; Phipps, D.; Kontopantelis, E.; Bower, P.; Campbell, S.; Haneef, R.; Avery, A. J.; Ashcroft, D. M., Prevalence, severity, and nature of preventable patient harm across medical care settings: systematic review and meta-analysis. BMJ 2019, 366, l4185.

Reviewer 2: Conclusion

Authors should add future directions. It also seemed important to me that the authors explore specific future directions, given their results on 'communication training.’

Response: Thank you for this comment. We added three sentences to stress the need for further communication trainings.

It reads: “Our research results indicate that the healthcare system would benefit from clear communication and mutual understanding, both in terms of satisfaction and reduction of preventable adverse events. Thus, we assume that it is beneficial and necessary to align approaches to train good communication. Therefore, we recommend the intro-duction of communication trainings for both sides, healthcare workers and patients, respective expectant mothers on a regular basis. Adding the important topic of communication to antenatal preparation classes could help to improve patient safety and satisfaction with birth. This is especially important for countries in which obstetric care needs to be further developed; however, the training approach at hand clearly needs to be adapted and re-evaluated in different cultural settings and in different languages, including countries with lower patient safety or issues regarding fertility rates. It is crucial to ensure that especially women who have language barriers or low-er communication skills and who are thus are at risk of not communicating safely can benefit from communication interventions to ensure safe childbirth and well-being of mothers as well as their offspring. Accordingly, communication training is recommended for different target groups and for improving diverse proximal or distal consequence.”

Round 2

Reviewer 1 Report

Thank you for replay

Author Response

Reviewer: 1. QCA reliability was not reported. For example, the inter-rater reliability in assigning the findings to categories.

Response: Thank you for this comment. We added a sentence about the process of finding consensus about the main categories and the respective answers.

It now reads: “Once the interviews had been transcribed verbatim, the research group (JD, FH, CD, MS; SL) including health psychologists, a Master of Science in health promotion and a sociologist worked out the main categories to answer the research questions. Several discussions were held until a consensus was reached for all categories. The results are presented in the next section and illustrated with original quotes.” (lines 257)

Reviewer: 2. The coding inductive/deductive is not transparent.

Response: To gain insights into the subjective perception of the effects of the communication lessons on the birth experience, we wanted to create an open narrative atmosphere. That is, our semi-structured interview guide covered the recall of situations (“please recall the whole process of delivery”), emotions (“what describes the feeling you associate with your birth experience”) and persons (“Which persons or professions were most important in providing or lacking support?”) in order to obtain a wide range and depth of narratives regarding the birth experience and perceived communication. From these responses we gathered expected (deductive) and unexpected (inductive) results without defining fixed answer categories beforehand. Unexpected results were, for example, the frequent mention of the learned empathy with the professional's work situation, which even led interviewees to feel stressed about having communicated well enough or the learning effect due to the expressed divergent needs of other training participants in the online classes. What was also unexpected for us was the difference between the time before the delivery room, when the women felt alone, and the reported care in the delivery room, when healthcare workers and partners were present. Interesting and not predefined as answering category was the different perception of the professionals. For some participants the midwife was the most important and actually sufficient caregiver, while others emphasized greater importance on a doctor's care. In conclusion, in our study, inductive and deductive categories are intertwined, as we asked about topics but did work with predefined answer categories.

In case you think this information should be included into the manuscript or the appendix, we are happy to make according changes. However, we did not do so to this point because we do not want to overburden the manuscript and the readers.

Reviewer: 3. The development of the guide is not very transparent. The principles of DOI or guide development are not referenced.

Response: Thank you for your question! The development of our interview guide based on our previous research and planning within the TeamBaby project, which is in detail described in the quoted study protocol. (Lippke, S., Wienert, J., Keller, F.M., Derksen, C., Welp, A., Kötting, L., Hofreuter-Gätgens, K., Müller, H., Louwen, F., Weigand, M. Ernst, K., Kraft, K., Reister, F., Polasik, A., Huener B., Seemann, B., Jennewein, l., Scholz, C. & Hannawa, A. (2019). Communication and patient safety in gynecology and obstetrics – study protocol of an intervention study. BMC Health Serv Res, 19(1), 908, doi:10.1186/s12913-019-4579-y).

Further, in previous interviews with obstetric healthcare workers we identified factors that facilitate or hinder safe communication from a professional perspective. (Schmiedhofer, M.; Derksen, C.; Keller, F. M.; Dietl, J. E.; Haussler, F.; Strametz, R.; Koester-Steinebach, I.; Lippke, S., Barriers and Facilitators of Safe Communication in Obstetrics: Results from Qualitative Interviews with Physicians, Midwives and Nurses. Int J Environ Res Public Health 2021, 18, (3)). Based on this data, we asked, for example, about the perceived support of the individual professional groups and their interaction and the extent to which women are able to express their own needs as those were reported as factors impacting patients’ safety.

In the evaluation of clinical birth data, associations were found between mothers' personal data and the risk of preventable adverse events. Based on this information, we specifically asked about the experience of medical complications and the perceived professional communication in the respective situation. (Hüner, B., Derksen, C., Schmiedhofer, M., Lippke, S., Janni, W., & Scholz, C. (2022). Preventable adverse events in obstetrics—systemic assessment of their incidence and linked risk factors. Healthcare, 10(1), 97. doi: 10.3390/healthcare10010097).

The examination of the health action process approach (HAPA) to improve communication and patient safety contributed to asking specifically for the aim of communicating well and the extent to which it was done from the interviewees’ perspective in recall. (Derksen, C., Kötting, L., Keller, F. M., Schmiedhofer, M., & Lippke, S. (2022). Psychological intervention to improve communication and patient safety in obstetrics: Examination of the health action process approach. Frontiers in psychology, 13, https://doi.org/10.3389/fpubh.2022.739100).

Finally, the first question (“Do you remember getting your first information about births in your life, and if yes from whom?”) was more used to stimulate the narrative flow and create a positive interview atmosphere than obtaining data to answer the research question.

To make the development of the interview guide more transparent, we added a sentence about the resources of our previous knowledge and planning, which now reads: “The semi-structured interview guide (table 2) includes main topics to stimulate a detailed description of the birth experience with a focus on communication, perceived support and the subjective impact of the communication training. It was flexibly adapted to the narrative flow and developed on the basis of our previous research results and project planning [19, 25, 32, 33].” (lines 210-214).

Reviewer: 4. Assignment of finding sites to research questions is usually done with ecological triangulation (e.g., according to Yin, 2014 and others)

Response: In this study, a qualitative content analysis approach was used, analyzing the word-by-word transcribed interviews against the background of the field notes to identify significant sequences according to the research questions. As outlined in the section to answer the question about the development of the interview guide, our theoretical and empirical previous work was used to develop the interview guide and to work out the results in broad discussions among the study group.

Reviewer: 5. The results are available in full reference, although here the frequencies per category per question should be reported.

Response: We focused our data analysis on the most reported issues, which we described and discussed in detail. The respondents put different emphases on their feelings, perceptions and evaluations, which cannot be solidly represented in quantitative categories by the evaluation method used. Therefore, our approach is to identify and compare the categories which are most relevant to answering the research question. Topics are highlighted with original citations to provide a concise and illustrative representation. Agreeing and disagreeing responses are also presented. Quantification would not make the results more precise because the themes emerged under different conditions in terms of complications or previous communicative expertise. The manuscript at hand covers the qualitative results of a mixed-methods approach, while the quantitative results will be published in another manuscript, which is currently under review and cited accordingly (30. Derksen, C.; Dietl, J. E.; Häussler, S.; Steinherr-Zazo, M.; Schmiedhofer, M.; Lippke, S., Communication training for pregnant women: Application of the Health Action P